# Functional Nanocomposites in the Development of Flexible Armor

**DOI:** 10.3390/ijms24065067

**Published:** 2023-03-07

**Authors:** Hassan Mahfuz, Vincent Lambert, Floria Clements

**Affiliations:** 1Department of Ocean and Mechanical Engineering, Florida Atlantic University, Boca Raton, FL 33431, USA; 2Technipfmc plc, Houston, TX 77044, USA; 3NextEra Energy Inc., Juno Beach, FL 33408, USA

**Keywords:** Flexible Armor, shear thickening fluid, nanocomposites, silane coupling agent, Gluta

## Abstract

The idea of flexible body armor has been around for many years. Initial development included shear thickening fluid (STF) as a backbone polymer to impregnate ballistic fibers such as Kevlar. At the core of the ballistic and spike resistance was the instantaneous rise in viscosity of STF during impact. Increase in viscosity was due to the hydroclustering of silica nanoparticles dispersed in polyethylene glycol (PEG) through a centrifuge and evaporation process. When STF composite was dry, hydroclustering was not possible due to absence of any fluidity in PEG. However, particles embedded within the polymer, covered the Kevlar fiber and offered some resistance to spike and ballistic penetration. The resistance was meagre and hence, the goal was to improve it further. This was achieved by creating chemical bonds between particles, and by strongly attaching particles to the fiber. PEG was replaced with silane (3-amino propyl trimethoxysilane), and a fixative cross-linker, Glutaraldehyde (Gluta), was added. Silane installed an amine functional group on the silica nanoparticle surface, and Gluta created strong bridges between distant pairs of amine groups. Amide functional groups present in Kevlar also interacted with Gluta and silane to form a secondary amine, allowing silica particles to attach to fiber. A network of amine bonding was also established across the particle-polymer-fiber system. In synthesizing the armor, silica nanoparticles were dispersed in a mixture of silane, ethanol, water, and Gluta, maintaining an appropriate ratio by weight, and using a sonication technique. Ethanol was used as a dispersion fluid and was evaporated later. Several layers of Kevlar fabric were then soaked with the admixture for about 24 h and dried in an oven. Armor composites were tested in a drop tower according to NIJ115 Standard using spikes. Kinetic energy at impact was calculated and normalized with the aerial density of the armor. NIJ tests revealed that normalized energy for 0-layer penetration increased from 10 J-cm^2^/g (STF composite) to 220 J-cm^2^/g for the new armor composite, indicating a 22-fold enhancement. SEM and FTIR studies confirmed that this high resistance to spike penetration was due to the formation of stronger C-N, C-H, and C=C-H stretches facilitated by the presence of silane and Gluta.

## 1. Introduction

Over the years, considerable attention has been given to the development of warrior systems that would provide mostly ballistic protection for individual soldier. These warrior systems, which are widely known as personnel armor, would mainly involve the protection of the torso and head. While protection of these body parts is of paramount importance, it is also true that a significant number of battlefield casualties and injuries would happen due to injuries inflicted on body extremities such as hands, arms, necks, and legs. Before the invention of penicillin, casualties from these injuries were the largest than any other categories during World Wars I and II. It is obvious that the protection of these body parts is important, and the armor systems to provide protection for such extremities must be very lightweight and flexible to allow maximum maneuverability of the soldier.

Body armors have been developed primarily based on Kevlar fibers in conjunction with ceramic and other protective layers to defeat high-speed projectiles or fragments. These body armors by virtue of their constituent materials and construction are not flexible enough to be used in extremities. Recent studies have shown that shear thickening behavior of a concentrated dispersion of fumed silica particles in PEG under steady shear can be successfully utilized for ballistic protection [1,2,3,4,5,6]. It has been demonstrated that composites made of Kevlar fabric reinforced by STF can offer equivalent low velocity ballistic performance similar to neat Kevlar, but with significantly fewer number of layers. Consequently, the resulting armor system is lighter and more flexible [7,8]. In STF, Shear-thickened state is characterized by the formation of transient clusters of colloidal or nano-sized particles. The formation of these flow-induced clusters results in an increased dissipation of energy and, consequently, the viscosity increases [9,10]. The effect can lead to a flow-induced solidification increasing the penetration resistance. 

In recent years, nanocomposites have emerged as new hybrid materials consisting of polymers and nanometer-sized inorganic particles. These nanocomposites have exhibited significant improvement in chemical, thermal, and mechanical properties with a very low loading of the inorganic particles, while still allowing conventional polymer processing. The benefit of nanoparticle infusion comes from the fact that the large amount of interphase zones in nanocomposites may serve as catalysts for prolific crack growth, creating a much greater number of new surfaces [11,12,13]. The creation of new surfaces can serve as an efficient mechanism to dissipate kinetic energy in the event of an impact. These interphase zones can also be visualized as defects. In nanocomposites, the density of these “defects” will be very high; the spacing between defects will approach interatomic distances [14]. As a result, a large fraction of atoms will sit very close to a defect. Any brittle crack developed in the material will be deflected and disseminated by these defects. Nanocomposites by nature can absorb energy during impact and, therefore, are appropriate to be introduced in the development of body armor. In addition to taking advantage of nanocomposites, current investigation took two new steps—one is to functionalize silica nanoparticles using a silane coupling agent, and the other is to introduce a fixative cross-linker, Glutaraldehyde (Gluta). Silane installed an amine functional group on the surface of silica particles, and Gluta facilitated forming a bridge connecting the functionalized particles and Kevlar. 

A systematic approach was undertaken in this investigation to develop flexible body armor using ballistic fibers such as Kevlar impregnated with a nanoparticle-polymer mix. The idea was to shift away from STF-based concept to a nanoparticle-polymer based armor to take advantage of nanocomposites properties. The novelty in the present approach was to use the polymer (Silane), not only to act as a host, but also to work as a coupling agent between particles and Kevlar in presence of water and a cross-linker. Amine functional groups on the surface of silica nanoparticles created a secondary amine reacting with Gluta and formed strong bonding across the particle-polymer-fiber system. 

## 2. Results and Discussion

Three types of characterizations, namely, FTIR, SEM, and NIJ Spike tests were performed in this study. These three characterizations were sufficient to evaluate the performance of the armor and explain the source of improvement in spike resistance both qualitatively and quantitatively. Ballistic tests were not performed in this investigation as it was not the scope of this paper. 

### 2.1. FTIR Studies 

FTIR spectra in the current investigation were generated using a Nicolet iS10 spectrometer. The absorbance shown in Figure 1 is ATR-FTIR spectra. All samples including Kevlar samples were directly placed on the ATR accessory, and data were collected and processed using the OMNIC spectral software. Quantity of samples (a few milligrams in weight) were kept same for all categories so that intensity of the peak absorbance of the functional groups could be compared. In Kevlar samples, fiber was added in small pieces with other ingredients. A background scan was acquired before collecting the sample data. Relative absorbance in Figure 1 was plotted by merging spectra from all four categories of samples so that their absorbance could be visualized relative to each other at identical frequencies. As such, the vertical axis was named as “Relative Absorbance”. Graphs were separated by 0.25 to 0.75 unit along the vertical axis, as seen in Figure 1, to better identify each category and visualize the peaks. The base for Kevlar + Silane + SiO_2_+Gluta sample was set at 0.0 in the vertical axis, as seen in Figure 1. 

FTIR spectra of silane, silica, and Kevlar after they are reacted with Gluta are shown in Figure 1. Relative absorbance with respect to frequency (wave number, cm^−1^) is plotted in Figure 1. It is observed that relative absorption is highest with (Kevlar + Silane + SiO_2_ + Gluta) sample @1140 cm^−1^ frequency, indicating an increase in the intensity of C-N stretch (secondary amine). High intensity suggests an increase in the amount of C-N functional groups per unit volume. The substantial increase in the C-N functional group is created due to Gluta interacting with amine functional groups of silica nanoparticles. The amine functional group on silica surface is formed due to the presence of silane and water in the admixture. The amide bond present in Kevlar also interact with Gluta, increasing the C-N functional groups. Such increase in secondary amine plays an important role in the bonding ability of silica nanoparticles to the Kevlar fabric. Without the presence of Kevlar, however, this intensity @1140 cm^−1^ is not high, as evident in the (Silane + SiO_2_ + Gluta) spectra. This indicates that amide functional groups of Kevlar bridging with silica particles via Gluta is responsible for this high intensity. Increase in C-H band and C-C stretch @944cm^−1^ and @1408cm^−1^ frequencies are also observed when compared with other samples. A distinct feature in (Kevlar + Gluta) sample is observed at @1710 frequency, which represents the weak bands overtones representing the carbonyl groups in Kevlar. This feature with weak intensity absorption is completely suppressed when silane is added, as seen in Figure 1. Reason for the disappearance of the carbonyl group is not known. One possibility is that, when silane is added, silated silica particles react with Kevlar in the presence of Gluta and form secondary amine (C-N stretch) that completely suppresses the carbonyl group. The characteristic @1408 cm^−1^ belongs to ring C=C-C stretching and bending vibrations, which is seen to have modest absorption across all samples, but highest in the (Kevlar + Silane + SiO_2_ + Gluta) sample. 

### 2.2. SEM Characterization

SEM examination was carried out using a JEOL 5800 SEM. Samples were fixed to the stage with double-sided carbon tape, and prior to mounting, they were coated with gold/palladium. 

The purpose of the chemical treatment of silica particles with silane and Gluta was to increase bonding of particles with themselves, as well as with Kevlar. The goal was to coat the fiber and cover interstitial spaces between fibers with silica particles as uniformly as possible. It was also important to examine particle dispersion and the amount of agglomerations in the polymer in presence of Gluta. Figure 2 shows the evolution of the bonding of silica particles with Kevlar fiber. Neat Kevlar fibers are shown in Figure 2a without any silica particles attached to them. When silica particles are dispersed in PEG (Figure 2b) without any silane coupling, the bonding is seen to be weak, and particles are agglomerated into large sizes. Not much of the interstitial spaces are covered either. When PEG is replaced with silane, and Gluta is added, particle dispersion is much more uniform, and silica particles densely adhere to fiber surfaces, as observed in Figure 2c. It is noticed that fiber is fully coated with fine silica particles, which can offer a formidable amount of resistance to spike penetration. This resistance is due to increased dynamic friction between the spike and strongly bonded silica particles. Interstitial spaces are also well-covered. If spike slips into this space without directly hitting the fiber, bonded silica particles would offer resistance to spike penetration. It is visible in Figure 2c that the size of agglomeration is small compared to that of Figure 2b. Longer sonication time helps disperse the particles more uniformly, reducing the agglomerations, while Gluta enforces attachment with Kevlar through secondary amine bonding. 

### 2.3. NIJ Spike Tests

After fabrication, armor composites were tested using a drop tower [15] following the NIJ (National Institute of Justice) Standard 0115.0., NIJ tests were performed with spikes having a drop weight of 2.0 kg, and drop heights ranging from 0.05 m to 2.5 m that produced impact energies from 1 J to 50 J, with an increment of 1 J. The velocity prior to impact was measured through a laser speed trap. Using the measured impact velocity, the total mass of the spike and drop mass, actual impact energy was calculated. During NIJ spike tests, energy level was increased at 1 J interval until there was penetration through the witness paper. Penetration depth was measured by the number of punctured witness paper placed underneath the fabric specimen, and in-between the consecutive neoprene sponge layers. After each impact, witness papers were inspected for tear and total puncture (hole). Only a hole in the paper was counted as a puncture. As energy level increased, the number of punctured witness papers increased to 1, 2, 3, etc., as shown in Figure 3. In the case when there was no penetration at all, such as with the Gluta system, energy level was increased until there was a failure of the backing material (polyethylene foam) placed underneath the neoprene sponge. The plot of impact energy versus penetration depth was used to compare the armor performance. Impact energy was normalized by areal density of the armor composite to eliminate the effect of any change in composite weight. Aerial density of each composite is indicated in Figure 3.

Results from spike tests are shown in Figure 3 for various armor composites. It is observed that energy required for 0-layer penetration (i.e., no penetration) for PEG-Silica (STF)/Kevlar composites (blue line) is about 10 J-cm^2^/g. When silica particles are silated and dispersed in PEG (PEG-Silane-Silica/Kevlar), the energy absorption level increases to 40 J-cm^2^/g, as indicated by the purple line. If PEG is replaced by silane, and Gluta is added (Silane-Silica-Gluta/Kevlar), 0-layer penetration energy goes as high as 220 J-cm^2^/g, indicated by the red line. This is 22 times higher than the STF-composite armor. Source of improvement in such penetration resistance is attributed to strong particle bonding with Kevlar facilitated by silane and Gluta due to the bridging of amino functional groups. 

## 3. Materials and Methods

### 3.1. Functionalized Silica Nanoparticles

In this approach, silica nanoparticles were functionalized with a silane coupling agent to enhance particle-polymer bonding and increase attachment of particles with the fiber. A silane coupling agent, Gelest-SIA0591.0 (Gelest Inc., East Steel Road, Morrisville, PA, USA), was used to functionalize silica particles. As shown in Figure 4, the silane coupling agent is made of a (CH_2_)_n_ linker, which is connected to an organofunctional group (R) and a silicon atom [16]. The silicon atom in turn is attached to three alkoxy (OH) groups. This alkoxy bond is hydrolytically unstable and, in presence of moisture, hydrolyses to an intermediate Si-OH bond, as shown. The Si-OH bond then condenses with the surface-bound OH groups present in the silica nanoparticles to form a stable Si-O-Si bond. It is important to note that the three terminal Si-OH groups can react with another molecule of silane because of higher mobility. However, due to the presence of silica nanoparticles, this mobility will be significantly interrupted. As terminal Si-OH intermediate bonds encounter silica particles, they will condense on the particle surface and form a more stable Si-O-Si bond, as shown in Figure 4. The R group in silane remains available for covalent reaction with the polymer. The separation between the organic functionality and the silicon is by three carbon atoms. The relatively shorter linker length causes the mixture of polymer and silica particles to be thermally more stable. Two types of silica particles were used in the investigation. One of them was nanoscale fumed silica of about 7 nm in diameter. The nanoscale fumed silica has a chain-like morphology. When dispersed into liquids, these chains bond together via weak hydrogen bonds forming a 3-D network. The 3-D chains can trap liquid and effectively increase viscosity at high shear rate. During the formation of fumed silica, hydroxyl groups become attached to silicon atoms on the particle surface. This makes silica surface hydrophobic and capable of hydrogen bonding with molecules of silane. The other category of silica particles used was relatively larger; about 20 nm in diameter and, unlike the previous kind, they were crystalline. These particles were procured from MTI Corporation, 860 Sout^h^ 19th Street, Richmond, CA 94804, USA. The reason for introducing larger silica particles was to improve dispersion and reduce the size of agglomeration. 

Functionalization of these two categories of particles was carried out following a relatively simple chemical treatment [18]. The silane coupling agent was aminopropyl trimethoxysilane, which was capable of coating 358 m^2^ of particle surface area per unit gm of silane. The amount of silane necessary for a batch of silica particles was calculated from the surface area of nanoparticles. The specific surface area of fumed silica (7 nm) was 390 m^2^/g, while that of 20 nm silica particle was 90.9 m^2^/g. The second category of silica nanoparticles was procured from Sigma-Aldrich. Particles were spherical in shape, and the diameter was between 10 nm and 20 nm. Purity was 99.5% based on trace metal analysis. Product and CAS numbers were 637,238 and 7631-86-9, respectively. The chemical treatment began with 95% ethanol and 5% water solution, while the required amount of silane was added through stirring into the mixture to yield a 2% final concentration of silane. Around five minutes were allowed for hydrolysis and silanol formation. In the next step, silica particles were stirred into the solution for about 2–3 min. Once the particles settled at the bottom of the beaker, the rest of the solution were decanted carefully. The particles were rinsed with ethanol and dried in an oven at 110 °C for about 10 min. Silated silica particles were then dispersed into a mixture of ethanol and polyethylene glycol (PEG). 

### 3.2. Armor-Composites Fabrication

Initially, the polymer in our case was polyethylene glycol (PEG). PEG is a condensation polymer of ethylene oxide and water with the chemical formula; HO − (CH2 − CH2 − O)_n_ − H. Subscript “n” indicates the average number of repeating oxyethylene groups varying from 4 to 80. We had used a low molecular weight (200) PEG, and it was soluble in water and ethanol. Hydroxyl group present in PEG allowed covalent coupling of PEG with silated silica nanoparticles. 

The fabric in the body armor was plain weave (style 706) Kevlar KM-2 (600 denier) procured from Hexcel Schwebel. Kevlar is a polyaromatic amide containing aromatic and amide groups which are oriented parallel to each other, providing a unique structure and rigidity. The individual polymer structure of Kevlar is held together by hydrogen bonds. The aromatic components have radial (spokes-like) orientation, which gives a high degree of symmetry and regularity to the internal structure of Kevlar. This crystalline-like regularity is the largest contributing factor for Kevlar’s impact strength.

A schematic view of the fabrication of Kevlar composites is shown in Figure 5. The manufacturing of the armor composites began with the dispersion of silated silica particles into a mixture of PEG and ethanol through a sonication process. Ethanol was added to aid in dispersion of silica particles and in soaking the Kevlar fabrics. The ratio of PEG to silica to Ethanol was approximately 1:1.22:13.17 by weight, respectively. This ratio resulted in a mixture of 45 wt% PEG and 55 wt% silica after drying out the ethanol. The dispersion of silica particles was carried out by ultrasonic cavitation, details of which are described in references [16,19]. After sonication for about an hour, the mixture was used to soak 12 layers of Kevlar fabric cut in dimensions of 30.48 cm × 38.10 cm. To impregnate the fabric, the sonicated mixture and the layers of fabric were placed in a plastic bag and sealed. After approximately 30 min of soaking, individual fabric layers were removed and laid flat in a furnace at 75 °C. Once the layers were dried, they were ready for spike tests. Areal densities for each category of fabric were measured by averaging the data from individual layer. 

### 3.3. Replacing PEG with Silane and Adding a Fixative Cross-Linker (Gluta)

Previous fabrication procedures included silated silica particles dispersed into a mixture of Ethanol and PEG. Silica particles were treated with silane first and then dispersed into PEG. Since silane by itself is a polymer, requirement of PEG as a polymer became redundant in the construction of the armor. In addition, a fixative cross-linker, namely, Glutaraldehyde (Gluta) was added to enhance particle-polymer bonding. Gluta was added according to the amount of silane present in the solution. It is a dialdehyde whose aldehydic groups are highly reactive and can form covalent bonds with functional groups such as amines and hydroxyl. Reaction of Gluta with amino groups in silica particles is shown in Figure 6. Gluta creates strong bridges between silica particles via di-amino groups present in silane. Such bridges are also formed with the fiber due to the presence of amino groups in Kevlar. These covalent bonds created by Gluta are stable, both mechanically and thermally. Mixture of Ethanol, silane, silica nanoparticles, and Gluta in appropriate concentration was homogenized and mixed through sonic cavitation [20]. After sonication for about three hours, the mixture was used to fabricate the armor composites, as described in the previous section.

## 4. Conclusions

A systematic approach to enhance spike resistance of Kevlar composites is investigated. It originates from the concept of STF composites, but deviates by eliminating PEG, and introducing relatively larger nanoscale (~20 nm) silica particles instead of fumed silica (~7 nm). Since particle size is large, van der Waals forces are reduced, and dispersion is more uniform. PEG is replaced by silane to develop N-H receptors on the surface of silica particles. A formaldehyde, known as Glutaraldehyde (Gluta), is added in the mix to bridge amine functional groups present in silica particles and in Kevlar fiber. Interaction of particles with silane and Gluta generates a secondary amine (C-N) bonding between particle-to-particle and particles-to-Kevlar. During impregnation, silica particles are bonded to Kevlar and fill up the interstitial spaces. It is important to note that the concentration of Si-containing active molecules is more in the solution compared to that of silica particles, and as such, there is a strong possibility of intermolecular reactions and the formation of chemical bonds as well. The resulting spike resistance of the new armor composites is thus phenomenal, about twenty-two times higher than that of STF composites. 

## Figures and Tables

**Figure 1 ijms-24-05067-f001:**
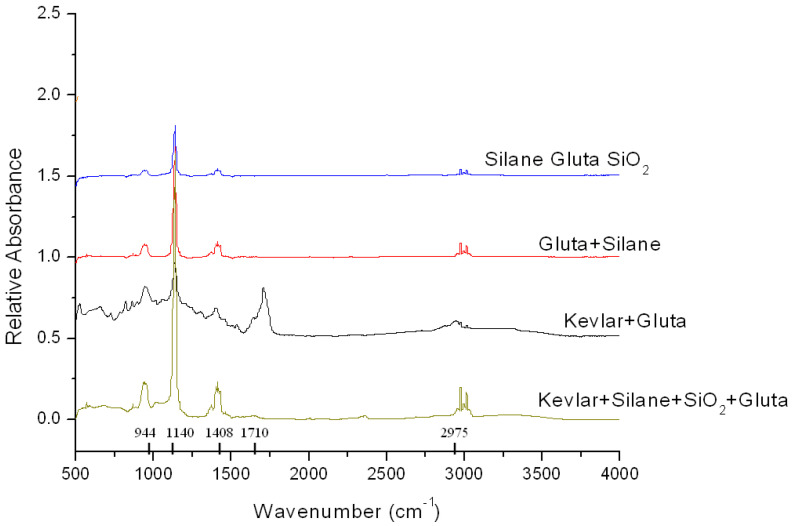
FTIR Spectra of Kevlar-Gluta Composites.

**Figure 2 ijms-24-05067-f002:**
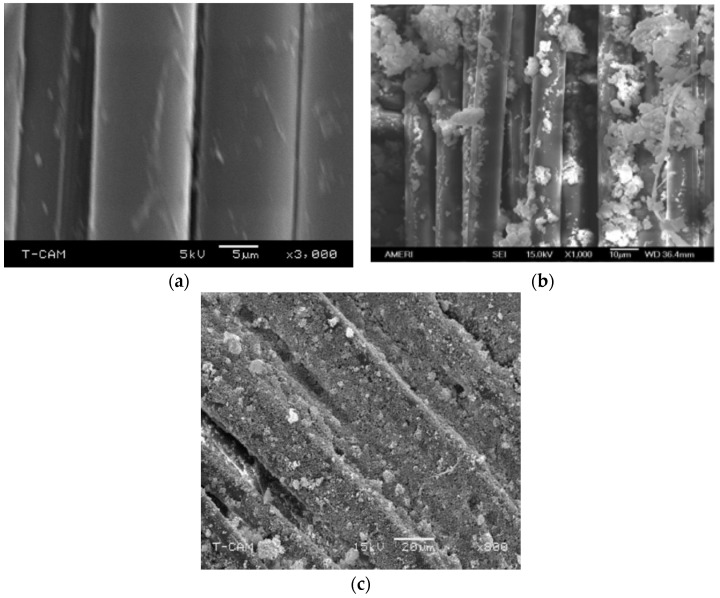
SEM micrographs of Kevlar composites: (**a**) Neat Kevlar, (**b**) Kevlar with PEG and Silica, and (**c**) Kevlar with Silane, Silica, and Gluta.

**Figure 3 ijms-24-05067-f003:**
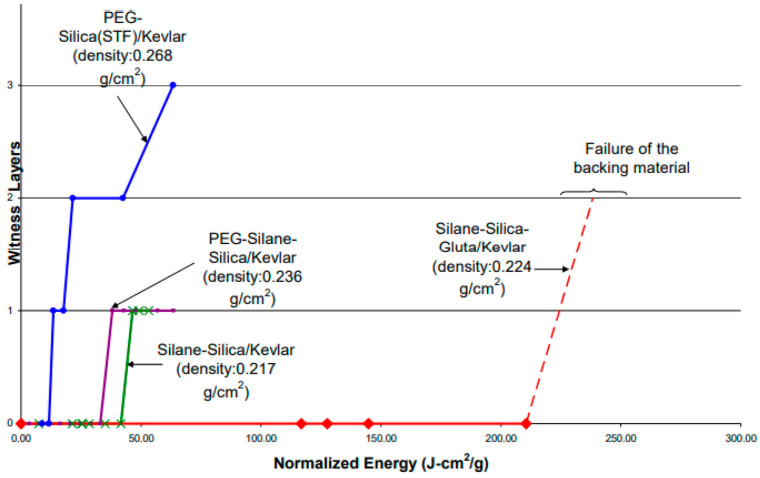
Spike Tests for Various Armor Composites.

**Figure 4 ijms-24-05067-f004:**
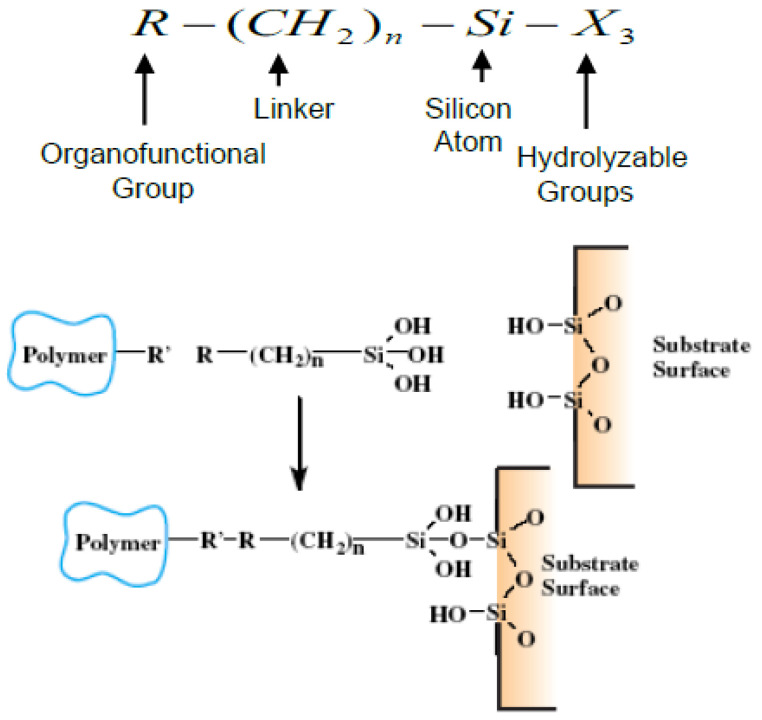
Silane coupling agent and functional groups [17].

**Figure 5 ijms-24-05067-f005:**
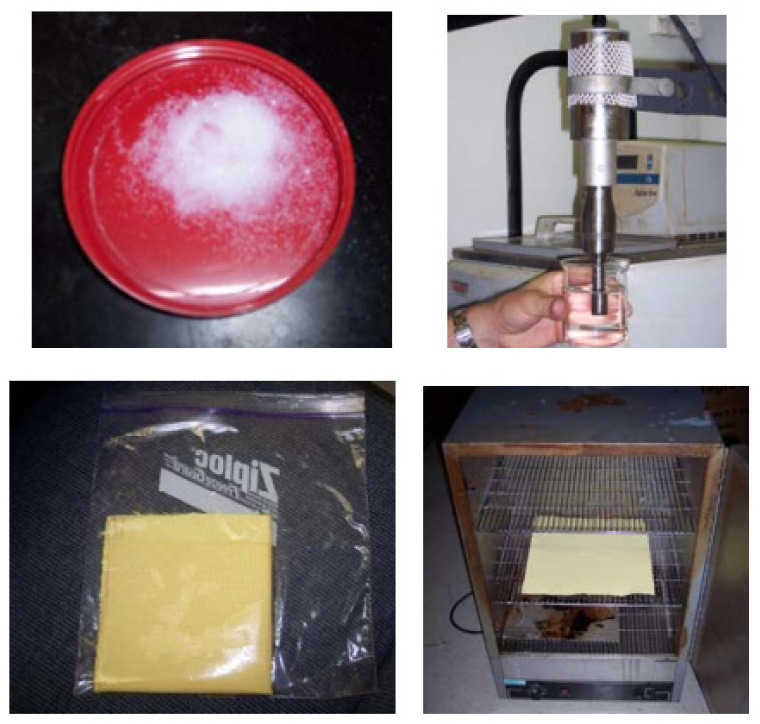
A schematic view of fabrication procedures.

**Figure 6 ijms-24-05067-f006:**
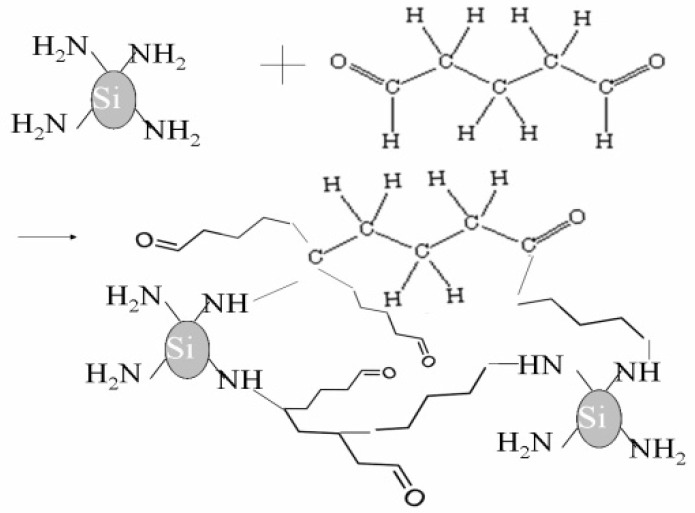
Cross-linking with Glutaraldehyde.

## Data Availability

Data are available in our published papers and theses.

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
