# Peer review of "Functional Nanocomposites in the Development of Flexible Armor"

_ijms, 2023, doi:10.3390/ijms24065067_

Round 1

Reviewer 1 Report

Dear Authors,

the submitted manuscript "Functional Nanocomposites in the Development of Flexible Armor" addresses the kevlar/silica composite for impact resistive composite material.

It was well designed and written for the proposed research purpose, thus recommend accept in IJMS as is.

Author Response

We would like to thank the reviewer for recommending to accept our paper as is.

Reviewer 2 Report

(i) The last paragraph of introduction should clearly address the novelty of work

(ii) The quality of figures 1, 2 & 5 need to be improved

(iii) The complete specifications (such as shape, dimensions and purity) of silica nano particles need to be

mentioned clearly

(iv) Only three types of characterizations were incorporated in the study. 

(v) Many other characterizations need to be incorporated in applications point of view. 

(vi) The quantity and quality of results need to be incorporated

The existing characterization may not be sufficient to conclude about the manuscript. 

Author Response

  1. The novelty in the present approach was to use the polymer (Silane) not only to act as a host but also to work as a coupling agent between particles and Kevlar in presence of water and a fixative cross-linker, Glutaraldehyde. Silane helped install an amine functional group on the surface of silica nanoparticles that created a secondary amine reacting with Gluta and formed bridges across the particle-polymer-fiber system. Strong bonds created by this approach offered significant resistance to spike penetration. As suggested by the reviewer, text has been added in the last paragraph of introduction in the revised manuscript.
  2. In our view quality of Figs 1 & 2 is okay. Figure 5 is changed to include only Kevlar systems and the scales are modified to be clearer.
  3. Silica nanoparticles were procured from Sigma Aldrich. Particles were spherical in shape and the diameter was between 10-20 nm. Purity was 99.5% based on trace metal analysis. Product and CAS numbers were 637238 and 7631-86-9, respectively. Text has been added accordingly.
  4. Yes, only 3 types of characterizations, namely, FTIR, SEM, and NIJ Spike Tests were included in the study.
  5. These 3 characterizations in our opinion were sufficient to evaluate the performance of the armor and explain the source of improvement in spike resistance both qualitatively and quantitatively. We agree with the reviewer that the application of the armor also includes ballistics, but that was not the scope of this paper. A paragraph is added at the beginning of the results and discussion section of the revised manuscript to clarify this point.
  6. It is a good question. It is to mention that during NIJ spike tests, energy level was increased at 1J interval until there was penetration through the witness paper. The penetration depth was measured by punctured witness paper placed underneath the fabric specimen and in-between consecutive neoprene sponge layers. After each impact, witness papers were inspected for tear and total puncture (hole). Only a hole in the paper was counted as a puncture. As energy level increased, number of punctured witness paper increased to 1, 2, 3, etc. as shown in Fig. 5. In the case when there was no penetration at all such as with the Gluta system, energy level was increased until there was a failure of the backing material (polyethylene foam) placed under the witness paper.  

Since our characterization was about spike penetration and about explaining the reasons for improvement using FTIR and SEM, we believe our conclusions are appropriate.  

Reviewer 3 Report

Dear Authors,
I found your manuscript (communication) interesting for potential readers and providing valuable knowledge about formatting new functionalized fibrous materials.
I have just one comment, regarding the reasons for providing quite fundamental information about mechanisms and reactions occurring when you prepared your research materials (lines 87-93 and 95-100). In my opinion that knowledge should be properly incorporated in the Introduction paragraph.
Regards!

Author Response

We very much appreciate reviewer’s comments about the fundamental information that we have provided in the Materials and Methods section of our paper. We believe those information belongs in that section. However as suggested by the reviewer, we have incorporated brief text in the introduction section as stated below:

In addition to taking advantage of nanocomposites, current investigation took two new steps – one is to functionalize silica nanoparticles using a silane coupling agent, and the other is to introduce a cross-linker, Glutaraldehyde (Gluta). Silane installed an amine functional group on the surface of silica particles and   Gluta facilitated forming a bridge connecting the functionalized particles, and Kevlar.  

Reviewer 4 Report

The paper presents an interesting approach to improving the ballistic properties of fabrics based on Kevlar. This kind of research, unfortunately, is currently particularly relevant for military applications, but no less relevant for the manufacture of protective equipment in the aerospace, auto industries, and building. The very principle of improving the interaction of the filler with the matrix is not new but can be extended to a wide class of objects.

The work has questions about the methodological part and design, which should be corrected:

1)  Line 18. Is it 3-amino propyl trimethoxysilane or something else?

2)  Line 100 and below. What do you mean by a silane? Is it a coupling agent or a hydrolysis product?

3)  How you controlled the condensation process? Hydrolyzed molecules of 3-amino propyl trimethoxysilane could more preferably react with each other due to their highest mobility in comparison to silica particles.

4)  Have you determined the relative rates of reactions and diffusion of these components?

5)  A detailed description of the linking mechanism shown in Fig. 1 is required.

6)  The data of IR spectroscopy are rather doubtful. How the measurements were made, reflection or transmission? How the data were normalized? Relative absorbance depends on the amount of sample and the correct band calibration must be performed for all samples. The description of IR method should be described in more detail or the conclusions should be reworked.

7)  1710 cm-1 corresponds to carbonyl groups in Kevlar, why it disappears in composite?

8)  1140 cm-1 also corresponds to C-O. Why do you think these are secondary amines?

9)  Figure 4a. Why are there so many particles on the original fiber? Is it dust?

10)              There are several typos in the text (skipping spaces between the number and the dimension, the beginning of sentences with a small letter, the symbol "0" instead of the degree symbol (°), Kevlar is a trademark and should be given by capital letter, etc.).

11)              Figure 5. Some text in Figure 5 is a blur.

Author Response

                               Response to Reviewer # 4

 1. Line 18. Is it 3-amino propyl trimethoxysilane or something else?

                      Yes, it is 3-amino propyl trimethoxysilane. Line 18 is corrected.

2. Line 100 and below. What do you mean by a silane? Is it a coupling agent or a hydrolysis  product?   

 It is a coupling agent as indicated in lines 82-83 and depicted in Fig. 1. Detailed description is in lines 84-91.  

3. How you controlled the condensation process? Hydrolyzed molecules of 3-amino propyl trimethoxysilane could more preferably react with each other due to their highest mobility in comparison to silica particles.

 It is a good question. We did not control the condensation process except controlling the concentration of silane, water, Gluta and silica nanoparticles in the mix. It is seen in Fig. 1 that silicon atom in the silane molecule is attached to the linker and to 3 hydrolysable alkoxy (OH) groups that form 3 Si-OH bonds in presence of water. These three terminal Si-OH group can react with another molecule of silane because of higher mobility as indicated by the reviewer. However, due to the presence of silica nanoparticles this mobility will be significantly interrupted. AS terminal Si-OH intermediate bonds encounter silica particles, they condense on the particle surface and form a more stable Si-O-Si bond as shown in Fig. 1.  

4. Have you determined the relative rates of reactions and diffusion of these components?

 No, we did not determine these rates as they were not within the scope of this paper.

5. A detailed description of the linking mechanism shown in Fig. 1 is required.

Description of linking mechanisms as shown in Fig. 1 is given in lines 85 – 94 of the manuscript.

6. The data of IR spectroscopy are rather doubtful. How the measurements were made, reflection or transmission? How the data were normalized? Relative absorbance depends on the amount of sample and the correct band calibration must be performed for all samples. The description of IR method should be described in more detail or the conclusions should be reworked.

 We respectfully disagree with the reviewer’s comments. Measurements were made through transmission. Samples were powders in pallet form. Amount of powder and thickness of the sample was identical for all categories. More description of IR method is included in section 3.1 as suggested by the reviewer.

7. 1710 cm-1 corresponds to carbonyl groups in Kevlar, why it disappears in composite?

We do not know the exact reason for the disappearance of the carbonyl group when silane is added in the Kevlar composite.  One possibility is that when silane is added, silated silica particles react with Kevlar in presence of Gluta and form the secondary amine (C-N stretch) suppressing the carbonyl group completely.   

8. 1140 cm-1 also corresponds to C-O. Why do you think these are secondary amines?

It seems to us that 1140 cm-1 does not correspond to C-O stretch. To our knowledge, C-O stretch appears in the region 1320 – 1210 cm-1 frequency. The secondary amine (C-N stretch) is clearly @ 1190 – 1130 cm-1 frequency [1, page 12]. Presence of N-H group in silane makes it more likely that C-N stretch will be formed rather than C-O stretch when interacting with Gluta.

9. Figure 4a. Why are there so many particles on the original fiber? Is it dust?

It is a typo. Fig 4a should be 4b and vice versa. Not sure how it got switched. Fig. 4a is for neat Kevlar and there are no particles there.

10. There are several typos in the text (skipping spaces between the number and the dimension, the beginning of sentences with a small letter, the symbol "0" instead of the degree symbol (°), Kevlar is a trademark and should be given by capital letter, etc.).

Temperature symbol and Kevlar spelling with upper case K are corrected in the revised manuscript.

11. Figure 5. Some text in Figure 5 is a blur.

A new version of Fig. 5 is included in the revised manuscript.

Reference:

  1. John Coates, “Interpretation of Infrared Spectra, A Practical Approach,” Encyclopedia of Analytical Chemistry, R.A. Meyers (Ed.), pp. 10815-10837, John Wiley & Sons Ltd, Chichester, 2000.

Round 2

Reviewer 2 Report

1. The revised manuscript seems to be not in MDPI format

2. In section 2.2, in fabrication section, better to explain with experimental photographs rather describing in sentences

3. Please check the term "details of which are described elsewhere [17-18]" in section 2.2

4. Ballistic impact test is the very important results required for this type of application, It is requested to authors to incorporate them with in the scope of this manuscript. In case of not including experimental results, at least with basic characterization, FEA results need to be incorporated.

Author Response

Response to Reviewer # 2 (Round 2)

  1. The revised manuscript seems to be not in MDPI format.

We agree. Both Word and PDF copies were submitted to the website, but it seems the file without MDPI format reached the reviewer. It is to note that, the revision for reviewer # 2 was marked with blue color in the revised manuscript. It is marked marked blue again in the 2nd revision.

  1. In section 2.2, in fabrication section, better to explain with experimental photographs rather describing in sentences.

A schematic view of fabrication steps is included in section 2.2 of the revised manuscript as Fig. 2.

Fig. 2: A schematic view of the fabrication procedures

  1. Please check the term "details of which are described elsewhere [17-18]" in section 2.2.

References [17] and [18] are changed.

  1. Ballistic impact test is the very important results required for this type of application, It is requested to authors to incorporate them with in the scope of this manuscript. In case of not including experimental results, at least with basic characterization, FEA results need to be incorporated.

We agree with the reviewer that ballistic tests are important for flexible armor. However, the entire effort in this project as funded by the army was to improve spike and stab resistance. Ballistic tests were not part of the project. FEA analysis of ballistic simulation using codes like LS-Dyna would be wonderful but that by itself would be another paper. Moreover, in the context of this “Communication” category paper, we believe FEA of ballistic impact is beyond the scope of this paper.

Reviewer 4 Report

The authors improved the Manuscript but I have some remarks:
1)           The new version of the Manuscript presents without an MDPI Template and has no line numbers, but I tried to find correspondent changes.
2)           The answer to the 4th previous question.  Due to the much more particle concentration of Si-containing active molecules with a comparison of silica particles, the intermolecular reactions look more possible. It would be interesting and useful to compare the probability of these processes. If the authors have not estimated that parameters correspond conclusions should be changed.
3)           Detail methods description must be given in the Experimental section (FTIR, SEM, Spike tests parameters and equipment). For example, it should be given to the manufacturer of the FTIR device. Also should be written the protocol for preparing samples the method of relative absorbance calculation and baseline creation if the authors compare the relative absorbance of the samples. How the powder was made from the samples with Kevlar fibers?

Author Response

Response to Reviewer # 4 (Round 2)

  • The new version of the Manuscript presents without an MDPI Template and has no line numbers, but I tried to find correspondent changes.

Changes in the manuscript for reviewer # 4 were marked with green color in the revised manuscript. It seems the information was not transmitted to the reviewer. In the next revision, the color for reviewer # 4 is green again.

  • The answer to the 4th previous question.  Due to the much more particle concentration of Si-containing active molecules with a comparison of silica particles, the intermolecular reactions look more possible. It would be interesting and useful to compare the probability of these processes. If the authors have not estimated that parameters correspond conclusions should be changed.

We are not sure whether intermolecular reactions are more likely to occur than the formation of siloxane (Si-O-Si) bond with silica particles. However, we agree with the reviewer that a study of this issue will be interesting. But within the limited scope of this “Communication” category paper, it is not practical at this time. Conclusion is therefore, modified to include high probability of this intermolecular reactions as suggested by the reviewer.

  • Detail methods description must be given in the Experimental section (FTIR, SEM, Spike tests parameters and equipment). For example, it should be given to the manufacturer of the FTIR device. Also should be written the protocol for preparing samples the method of relative absorbance calculation and baseline creation if the authors compare the relative absorbance of the samples. How the powder was made from the samples with Kevlar fibers?

FTIR: The spectra were generated using a Nicolet iS10 FTIR spectrometer. The absorbance shown in Fig. 4 (revised manuscript) is ATR-FTIR spectra. All samples including Kevlar samples were directly placed on the ATR accessory and data were collected and processed using the OMNIC spectral software. A background scan was acquired before collecting the sample data. Relative absorbance in Fig. 4 was plotted by merging spectra from all 4 categories of samples so that their absorbance could be compared relative to each other at identical frequencies. As such, the vertical axis was named as “Relative Absorbance.” Graphs were separated by 0.25 to 0.75 unit along the vertical axis as seen in Fig. 3 to better identify each category and visualize the peaks. The base for Kevlar+Silane+SiO2+Gluta sample was set at 0.0 in the vertical axis as seen in the figure.  

SEM: SEM examinations were carried out using a JEOL 5800 SEM. Samples were fixed to the stage with double-sided carbon tape, and prior to mounting, they were coated with gold/palladium. Gold/palladium coating prevented charge build-up by the electrons absorbed by the specimen. It gave the non-conductive specimen electrical conductivity to reduce the ability to attain an electrostatic charge. This enabled the use of higher voltage to increase clarity at a higher magnification.

Spike Test: We believe in the context of a “Communication” category paper, we have provided sufficient information in section 3.3 about the spike test. A complete description of the drop tower can be found in Reference [20] as indicated in the manuscript.  

Round 3

Reviewer 2 Report

The manuscript can be accepted

Author Response

We like to thank the reviewer for accepting our paper.

Reviewer 4 Report

1)      The commonly known information about the methods descriptions should be removed, especially for a “Communication” category paper.

Here are that examples from the text:

(page 5.) "FTIR is most useful for identifying chemical bonds that are either organic or inorganic. And its absorption spectrum is almost like molecular fingerprint. The change in infrared (IR)absorption is related with intrinsic vibrational modes of molecules. Each chemical functional group is associated with its own set of vibrational modes with their corresponding frequencies of vibration. "

(page 6.) "Gold/palladium coating prevented charge build-up by the electrons absorbed in the specimen. It gave the non-conductive specimen electrical conductivity to reduce the ability to attain an electrostatic charge. This enabled the use of higher voltage to increase clarity at a higher magnification."

2)      Previously you wrote: “Measurements were made through transmission. Samples were powders in pallet form. The amount of powder and thickness of the sample was identical for all categories.

2.1) Which pallet thicknesses were in that experiment?

2.2) How the high-strength Kevlar fiber was powdered into the powder?

2.3) You have written common information without the important experimental conditions.

3) The FTIR experimental data discussion looks incorrect. Judging by the data (different intensities for all main groups, with approximately the same ratio of the intensities of the main peaks for each of the samples), the relative absorbance changes due to a different amount of the sample or some other factor. Therefore, some conclusions indicated by the authors based on a comparison of the intensities of any groups for different samples do not look reliable. This indirectly confirms the "disappearance" of the carbonyl group.

Author Response

Response to Reviewer # 4 (Round 3)

  • The commonly known information about the methods descriptions should be removed, especially for a “Communication” category paper.

Here are that examples from the text:

(page 5.) "FTIR is most useful for identifying chemical bonds that are either organic or inorganic. And its absorption spectrum is almost like molecular fingerprint. The change in infrared (IR)absorption is related with intrinsic vibrational modes of molecules. Each chemical functional group is associated with its own set of vibrational modes with their corresponding frequencies of vibration. "

(page 6.) "Gold/palladium coating prevented charge build-up by the electrons absorbed in the specimen. It gave the non-conductive specimen electrical conductivity to reduce the ability to attain an electrostatic charge. This enabled the use of higher voltage to increase clarity at a higher magnification."

Commonly known information is removed as suggested by the reviewer.

  • Previously you wrote: “Measurements were made through transmission. Samples were powders in pallet form. The amount of powder and thickness of the sample was identical for all categories.”

2.1) Which pallet thicknesses were in that experiment?

2.2) How the high-strength Kevlar fiber was powdered into the powder?

2.3) You have written common information without the important experimental conditions.

The statement about samples was corrected in revision-2 of the manuscript. We reached out to the Lab where FTIR experiment was conducted, and we like to add that quantity of samples (a few milligrams in weight) were kept same for all categories so that intensity of the peak absorbance of the functional groups could be compared. Please note that in Kevlar samples, fiber was added in small piece with other ingredients. Text is added in section 3.1 (marked Green) to reflect the changes.

  • The FTIR experimental data discussion looks incorrect. Judging by the data (different intensities for all main groups, with approximately the same ratio of the intensities of the main peaks for each of the samples), the relative absorbance changes due to a different amount of the sample or some other factor. Therefore, some conclusions indicated by the authors based on a comparison of the intensities of any groups for different samples do not look reliable. This indirectly confirms the "disappearance" of the carbonyl group.

We respectfully disagree with the reviewer’s assertion that FTIR experimental data discussion looks incorrect. We stand by our FTIR experiment and interpretation of data.
